# Novel Red Light-Absorbing Organic Dyes Based on Indolo[3,2-b]carbazole as the Donor Applied in Co-Sensitizer-Free Dye-Sensitized Solar Cells

**DOI:** 10.3390/ma14071716

**Published:** 2021-03-31

**Authors:** Zhanhai Xiao, Bing Chen, Xudong Cheng

**Affiliations:** 1State Key Laboratory of Advanced Technology for Materials Synthesis and Processing, Wuhan University of Technology, Wuhan 430070, China; xiaozhanhai@whut.edu.cn; 2State Key Laboratory of Magnetic Resonance and Atomic and Molecular Physics, Innovation Academy for Precision Measurement Science and Technology, Chinese Academy of Science, Wuhan 430071, China

**Keywords:** organic dyes, indolo[3,2-b]carbazole, DFT calculations, dye aggregation, benzothiadiazole

## Abstract

Three novel organic dyes (**D6**, **D7** and **D8**), based on indolo[3,2-b]carbazole as the donor and different types of electron-withdrawing groups as the acceptors, were synthesized and successfully applied in dye-sensitized solar cells (DSSCs). Their molecular structures were fully characterized by ^1^H NMR, ^13^C NMR and mass spectroscopy. The density functional theory (DFT) calculations, electrochemical impedance spectroscopy analysis, UV–Vis absorption characterization and tests of the solar cells were used to investigate the photophysical/electrochemical properties as well as DSSCs’ performances based on the dyes. Dye **D8** showed the broadest light-response range (300–770 nm) in the incident monochromatic photo-to-electron conversion efficiency (IPCE) curve, due to its narrow bandgap (1.95 eV). However, dye **D6** exhibited the best device performance among the three dyes, with power conversion efficiency of 5.41%, J_sc_ of 12.55 mA cm^−2^, V_oc_ of 745 mV and fill factor (FF) of 0.59. We also found that dye aggregation was efficiently suppressed by the introduction of alkylated indolo[3,2-b]carbazole, and, hence, better power conversion efficiencies were observed for all the three dyes, compared to the devices of co-sensitization with chenodeoxycholic acid (CDCA). It was unnecessary to add adsorbents to suppress the dye aggregation.

## 1. Introduction

In the past three decades, scientists have paid much attention to the dye-sensitized solar cells (DSSCs). They have been regarded as an alternative energy source since the first report in 1991, due to their ability to convert solar energy to electricity at a low cost and excellent photovoltaic performance [1,2,3,4]. To date, Ru-complex and Zn-porphyrin sensitizers have achieved high power conversion efficiencies (PCEs) of 11.9% and 13.5%, respectively [5,6]. However, the high cost, limited resource of ruthenium and the complex synthesis procedure of zinc porphyrin impede their further application in DSSCs. Therefore, worldwide scientists have been involved in developing metal-free organic dyes with high PCE, due to their flexible structural modification, simple synthesis, low toxicity and low cost [7,8]. Recently, DSSCs based on metal-free organic dyes have achieved a very high PCE of 14% [9]. However, those dyes have complex structures and are difficult to synthesize. In addition, they need particular electrolytes and a co-sensitizer to improve open circuit voltage and suppress the dye aggregation to achieve high PCE. These reasons work against the commercialization of dyes. Therefore, the organic dyes featuring simple synthesis and without the necessity for using a co-sensitizer to improve PCE in the DSSCs could be a promising research field.

Among various metal-free organic dyes, the dyes with donor–π–bridge–acceptor (D–π–A) structures exhibit outstanding performance with simple molecular structures. For the typical D–π–A dyes, the structural modifications of the donor and acceptor can effectively tune the energy levels and improve intramolecular charge transfer (ICT) from D to A, therefore providing a direct way for acquiring high PCE [10]. Many kinds of promising donors have been reported, such as carbazole [11,12,13,14,15], di(1-benzothieno)[3,2-b:2′,3′-d]pyrrole [16,17], triphenylamine [18], dithieno[3,2-b]pyrrolobenzotriazole [19], and so on. Recently, indolo[3,2-b]carbazole featuring a larger conjugated plane than triphenylamine and carbazole has been reported as an excellent donor to construct organic optoelectronic functional materials, due to its outstanding hole-donating ability [20,21]. With its high thermal and chemical stability, indolo[3,2-b]carbazole was proven to be a good candidate as the D unit of organic dyes [22]. We previously reported a series of efficient D–D–π–A organic dyes based on indolo[3,2-b]carbazole as the first donor [23,24], triphenylamine as the second donor, and thiophene cyanoacetic acid as the acceptor, showing efficient photovoltaic performance with a highest PCE up to 6.34%. That study indicated that indolo[3,2-b]carbazole was a potential donor group in D–D–π–A-type organic dyes. However, it was rarely reported that indolo[3,2-b]carbazole was used as a single donor group. The study focus on the behaviors of indolo[3,2-b]carbazole-based D–π–A and D–A–π–A dyes is also an interesting field. In addition, the light-response ranges based on previously reported dyes were not satisfactory, because they can only absorb the visible light, which leads to comparably low PCE, compared with those near-infrared-absorbing organic dyes. To expand the light absorption to longer wavelengths, an excellent acceptor should be introduced to the dyes to enhance the D–π–A effects. We found that 2,1,3-benzothiadiazole as promising electron acceptor plays a crucial role in power conversion efficiency [25,26]. Because of its strong electron withdrawing ability, benzothiadiazole could lower the energy level of the lowest unoccupied molecular orbital (LUMO) and further reduce the energy gap of sensitizers [27,28]. Therefore, benzothiadiazole may be a suitable acceptor and is easily connected by some well-performed donors to enhance dye performances with simple dye structures.

In this work, we synthetized three dyes (**D6**, **D7** and **D8**) by adopting indolo[3,2-b]carbazole as single donor to simplify synthesis and thiophene cyanoacetic acid, benzothiadiazole benzoic acid and benzothiadiazole thiophene cyanoacetic acid as respective acceptors to modify the bandgap of dyes. The structures of **D6**--**D8** were shown in Figure 1. As the results show, the introduction of benzothiadiazole effectively expanded the light absorption of dyes to 690 nm. In addition, the dye aggregation was also suppressed by the two alkyl chains of indolo[3,2-b]carbazole, demonstrating better PCE even in the co-sensitizer-free DSSCs. Finally, the systematic study of their photophysical and electrochemical properties, as well as the resulting DSSCs performances based on the three dyes, was also reported.

## 2. Experimental

### 2.1. Materials and Methods

All the chemicals used in this study were purchased from Sigma–Aldrich (Saint Louis, MO, USA) and J&K Chemical Ltd (Beijing, China). The solvents used in synthesis were purified using standard processes. The reactions in this study were carried out in the N_2_ atmosphere. The separation of the compounds was carried out on column chromatography using silica gel (200–300 mesh).

Nuclear Magnetic Resonance (NMR) spectra (^1^H and ^13^C) characterization were tested on a Bruker 500 MHz spectrometer in CDCl_3_ and DMSO-*d*_6_. Mass spectra were carried out on a Bruker microTOF-Q. The absorption spectra of dyes were recorded on a HP 8453 spectrophotometer. Electrochemical properties were studied by cyclic voltammetry measurements on a CHI604D electrochemical workstation, using Bu_4_NPF_6_ as supporting electrolyte, dry dichloromethane as solvent and N_2_ as protection gas. The scan rate was 50 mV s^−1^. The ferrocene/ferrocenium (Fc/Fc^+^) redox couple was used as the internal standard. The current voltage (J–V) tests of the DSSCs were carried out on Keithley 2400 source meter under simulated AM 1.5 G (100 mW cm^−2^) illumination with a solar light simulator. Incident monochromatic photo-to-electron conversion efficiency (IPCE) was measured from 300 to 800 nm by a Spectral Products DK240 monochromator. The electrochemical impedance spectra (EIS) were carried out on a CHI604D electrochemical work station under dark conditions.

### 2.2. Fabrication and Characterization of DSSCs

Nanocrystalline TiO_2_ films were prepared by screen printing two kinds of TiO_2_ nanoparticles (a 12 μm nanoporous layer and a 4 μm scattering layer) over the conductive side of the conducting glass. The active area of the TiO_2_ was 0.196 cm^2^. The photoanode was prepared by immersing the TiO_2_ film into a 0.3 mM dye solution in THF for 24 h under the dark. After the dye adsorption, the photoanode was washed with dichloromethane (DCM) and then dried. The electrolyte contained acetonitrile–valeronitrile (85:15, v/v), composed of 0.6 M BMII (1-butyl-3-methylimidazolium iodide), 0.05 M LiI, 0.03 M I_2_, 0.5 M 4-tert-butylpyridine and 0.1 M guanidinium thiocyanate. The electrolyte was inserted into the interspace between the photoanode and cathode from the two holes predrilled on the back of the counter electrode. At last, the holes were sealed with a Surlyn film and a thin glass.

### 2.3. Synthesis

Synthetic routes to the three dyes **D6**, **D7** and **D8** are depicted in Scheme 1. Compound **1** was prepared via substitution reaction of 5,11-Dihydroindolo[3,2-b]carbazole with 2-ethylhexyl bromide and NaOH in DMSO, according to the literature [29]. Compound **1** was prepared by controlling the amount of N-bromosuccinimde (NBS) to provide monobrominated compound **2**. Then, key intermediate compounds, such as pinacol boronic ester 2, were afforded by coupling bis(pinacolato)diboron with compound **3**. Compounds **5**, **6**, **3a**, **3b** and **3c** were synthesized by Suzuki coupling reaction. The dyes **D6** and **D8** were synthesized by condensation of aldehydes **3a** and **3c** with piperidine cyanoacetic acid. Dye **D7** was synthesized by hydrolysis of compound **3b** in sodium hydroxide solution. The NMR and mass spectra of **D6**–**D8** are shown in Appendix A.


**2-bromo-5,11-bis(2-ethylhexyl)-5,11-dihydroindolo[3,2-b]carbazole (2)**


In a 100 mL flask, compound **1** (2.3 g, 4.8 mmol) was dissolved in 40 mL THF solution and stirred at 0 °C for 15 min. Under light-shielding conditions, (937 mg, 5.2 mmol) N-bromosuccinimide (NBS) was added portion-wise. After 2 h of reaction, the temperature was gradually raised to room temperature, and the reaction was continued for 24 h. The reaction mixture was poured into water and extracted with DCM, and the combined extracts were washed with water and dried with anhydrous NaSO_4_. The solvent was removed with a rotary evaporator, and the residue was isolated by silica gel column chromatography with petroleum ether (PE) as eluent to afford compound **2** as a yellow–green solid (yield: 81%, 2.17 g). ^1^H NMR (CDCl_3_, 600 MHz, ppm): δ 8.27 (d, *J* = 1.8 Hz, 1H), 8.19 (d, *J* = 7.8 Hz, 1H), 7.92 (s, 1H), 7.83 (s, 1H), 7.53−7.48 (m, 2H), 7.38 (d, *J* = 8.4 Hz,1H), 7.25–7.21 (m, 2H), 4.19–4.12 (m, 4H), 2.17–2.12 (m, 2H), 1.48–1.29 (m, 16H), 0.97–0.90 (m, 12H). ^13^C NMR (CDCl_3_, 150 MHz, ppm): δ 142.37, 140.76, 140.74, 136.72, 128.11, 126.03, 124.65, 123.43, 122.79, 122.71, 121.54, 120.25, 118.09, 110.54, 110.11, 108.84, 99.12, 99.01, 47.83, 47.81, 39.40, 39.34, 31.20, 31.16, 28.97, 28.94, 24.64, 23.26, 23.21, 14.24, 14.22, 11.14, 11.12. HRMS (ESI, m/z): [M]^+^ calcd for [C_34_H_43_BrN_2_]^+^: 558.2610, found: 558.2630.


**5,11-bis(2-ethylhexyl)-2-(4,4,5,5-tetramethyl-1,3,2-dioxaborolan-2-yl)-5,11-dihydroindolo[3,2-b]carbazole (3)**


In a 250 mL flask, compound **2** (2 g, 3.57 mmol), bis(pinacolato)diboron (B_2_pin_2_) (2.72 g, 10.72 mmol), anhydrous potassium acetate (KOAc) (1.05 g, 10.72 mmol) and dichloro[1,1′-bis(diphenylphosphino)-ferrocene]palladium(II) (Pd(dppf)Cl_2_) (261 mg, 0.357 mmol) were dissolved in 100 mL degassed dioxane under N_2_ atmosphere. The reaction was stirred at 80 °C for 24 h. Then, water was added and extracted with DCM. The organic layer was combined and dried with anhydrous NaSO_4_. The solvent was removed under vacuum, and the crude compound was purified by column chromatography with PE:DCM = 1:1 as the eluent to give brown viscous oil compound **3** (yield: 69%, 1.50 g). ^1^H NMR (CDCl_3_, 500 MHz, ppm): δ 8.70 (s, 1H), 8.20 (d, *J* = 7.7 Hz, 1H), 8.07 (s, 1H), 7.99 (s, 1H), 7.95 (d, *J* = 8.5 Hz, 1H), 7.49–7.46 (m, 1H), 7.42–7.38 (m, 2H), 7.25–7.22 (m, 1H), 4.33–4.23 (m, 4H), 2.24–2.18 (m, 2H), 1.43 (s, 12H), 1.41–1.26 (m, 16H), 0.97–0.94 (m, 6H), 0.91–0.88 (m, 6H). ^13^C NMR (CDCl_3_, 125 MHz, ppm): 144.40, 142.13, 136.83, 136.45, 134.82, 132.25, 131.34, 127.76, 127.45, 125.61, 122.83, 120.04, 117.91, 108.72, 108.09, 99.33, 98.93, 83.62, 47.72, 39.21, 31.02, 28.84, 25.05, 24.48, 23.23, 14.11, 11.05. HRMS (ESI, m/z): [M]^+^ Calcd. for (C_40_H_55_BN_2_O_2_), 606.4357, found: 606.4369.


**methyl 4-(7-bromobenzo[c][1,2,5]thiadiazol-4-yl)benzoate (5)**


(4-(methoxycarbonyl)phenyl)boronic acid (1 g, 5.5 mmol), 4,7-dibromobenzo[c][1,2,5]thiadiazole (1.96 g, 6.6 mmol), Pd(PPh_3_)_4_ (321 mg, 0.27 mmol), THF (150 mL) and 2 M K_2_CO_3_ (50 mL) were added in a 250 mL flask under N_2_ atmosphere. After stirring at 80 °C for 8 h, the mixture was extracted with DCM and removed under reduced pressure. The residue was isolated by silica gel column chromatography with PE:DCM = 1:1 as the eluent to give faint yellow solid compound **5** (780 mg, 41%). ^1^H NMR (CDCl_3_, 500 MHz, ppm): 8.21 (d, *J* = 8.4 Hz, 2H), 8.00 (d, *J* = 8.0 Hz, 2H), 7.97 (d, *J* = 7.2 Hz, 1H), 7.65 (d, *J* = 7.6 Hz, 1H), 3.98 (s, 3H). ^13^C NMR (CDCl_3_, 125 MHz, ppm): 166.77, 153.91, 152.88, 140.94, 132.84, 132.19, 130.12, 129.95, 129.17, 128.79, 114.25, 52.31. HRMS (ESI, m/z): [M + H]^+^ Calcd. for (C_14_H_10_BrN_2_O_2_S), 348.9646, found: 348.9651.


**5-(7-bromobenzo[c][1,2,5]thiadiazol-4-yl)thiophene-2-carbaldehyde (6)**


Compound **4** (0.5 g, 3.2 mmol), 4,7-dibromobenzo[c][1,2,5]thiadiazole (1.41 g, 4.8 mmol), Pd(PPh_3_)_4_ (321 mg, 0.27 mmol), THF (50 mL) and 2 M K_2_CO_3_ (50 mL) were added in a 100 mL flask under N_2_ atmosphere. The reaction solution was refluxed for 12 h. The resulting mixture was extracted with DCM and washed with distilled water. The organic layer was dried with anhydrous NaSO_4_. The solvent was removed under vacuum, and the crude compound was purified by column chromatography with PE:DCM = 1:1 as the eluent to give compound **6** (yield: 35%, 364 mg). ^1^H NMR (CDCl_3_, 500 MHz, ppm): 10.03 (s, 1H), 8.21 (d, *J* = 4.2 Hz, 1H), 7.96 (d, *J* = 7.8 Hz, 1H), 7.88 (d, *J* = 4.1 Hz, 1H), 7.86 (d, *J* = 7.5 Hz, 1H), ^13^C NMR (CDCl_3_, 125 MHz, ppm): 182.99, 147.56, 143.95, 136.67, 134.05, 132.14, 131.35, 128.54, 127.37, 125.75, 115.11. HRMS (ESI, m/z): [M]^+^ Calcd. for (C_11_H_5_BrN_2_OS_2_), 323.9027, found: 323.9011.


**5-(5,11-bis(2-ethylhexyl)-5,11-dihydroindolo[3,2-b]carbazol-2-yl)thiophene-2-carbaldehyde (3a)**


Compound **3** (300 mg, 0.49 mmol), compound **4** (92 mg, 0.59 mmol), Pd(PPh_3_)_4_ (46 mg, 0.04 mmol), THF (30 mL) and 2 M K_2_CO_3_ (1 mL) were added in a 100 mL flask under N_2_ atmosphere. The reaction solution was refluxed for 12 h. The resulting mixture was extracted with DCM and washed with distilled water. The organic layer was dried with anhydrous NaSO_4_. The solvent was removed under vacuum, and the crude compound was purified by column chromatography with PE:DCM = 1:1 as the eluent to give compound **3a** (yield: 51%, 149 mg). ^1^H NMR (CDCl_3_, 500 MHz, ppm): 9.89 (s, 1H), 8.47 (s, 1H), 8.20 (d, *J* = 7.80 Hz, 1H), 7.99 (s, 2H), 7.76–7.80 (m, 2H), 7.52–7.46 (m, 2H), 7.25–7.22 (m, 2H), 4.40–4.32 (m, 4H), 2.18–2.08 (m, 2H), 1.41–1.26 (m, 16H), 0.91–0.86 (m, 6H), 0.80–0.75 (m, 6H). ^13^C NMR (CDCl_3_, 125 MHz, ppm): 183.13, 164.45, 155.45, 148.32, 145.16, 142.36, 142.09, 136.95, 133.45, 132.35,126.71, 124.66, 123.73, 123.06, 122.89, 122.17, 121.03, 118.11, 110.34, 109.63, 99.73, 47.43, 38.99, 30.59, 28.34, 24.37, 23.45, 14.33, 11.12. HRMS (ESI, m/z): [M]^+^ Calcd. for (C_39_H_46_N_2_OS), 590.3331, found: 590.3316.


**methyl 4-(7-(5,11-bis(2-ethylhexyl)-5,11-dihydroindolo[3,2-b]carbazol-2-yl)benzo[c][1,2,5]thiadiazol-4-yl)benzoate (3b)**


Compound **3** (300 mg, 0.49 mmol), compound **5** (207 mg, 0.59 mmol), Pd(PPh_3_)_4_ (46 mg, 0.04 mmol), THF (30 mL) and 2 M K_2_CO_3_ (1 mL) were added in a 100 mL flask under N_2_ atmosphere. The reaction solution was refluxed for 12 h. The resulting mixture was extracted with DCM and washed with distilled water. The organic layer was dried with anhydrous NaSO_4_. The solvent was removed under vacuum, and the crude compound was purified by column chromatography with PE:DCM = 1:1 as the eluent to give compound **3b** (yield: 39%, 144 mg). ^1^H NMR (CDCl_3_, 500 MHz, ppm): 8.95 (s, 1H), 8.37 (s, 1H), 8.31 (s, 1H), 8.28 (d, *J* = 7.56 Hz, 1H), 8.22 (d, *J* = 8.50 Hz, 2H), 8.18 (d, *J* = 8.50 Hz, 2H), 8.13 (d, *J* = 8.40 Hz, 2H), 8.08 (s, 2H), 7.68 (d, *J* = 8.66 Hz, 2H), 7.52 (d, *J* = 8.26 Hz, 1H), 7.47–7.43 (m, 1H), 7.21–7.16 (m, 1H), 4.42–4.34 (m, 4H), 3.90 (s, 3H), 2.20–2.09 (m, 2H), 1.44–1.16 (m, 16H), 0.93–0.73 (m, 12H). ^13^C NMR (CDCl_3_, 125 MHz, ppm): 166.71, 154.23, 153.82, 142.10, 136.82, 136.71, 134.97, 130.04, 129.84, 128.04, 127.11, 123.02, 122.80, 122.67, 109.50, 100.23, 52.73, 47.33, 35.58, 31.72, 30.64, 29.43, 28.44, 24.22, 23.01, 14.33, 11.27. HRMS (ESI, m/z): [M]^+^ Calcd. for (C_48_H_52_N_4_O_2_S), 748.3811, found: 748.3826.


**5-(7-(5,11-bis(2-ethylhexyl)-5,11-dihydroindolo[3,2-b]carbazol-2-yl)benzo[c][1,2,5]thiadiazol-4-yl)thiophene-2-carbaldehyde (3c)**


Compound **3** (300 mg, 0.49 mmol), compound **6** (193 mg, 0.59 mmol), Pd(PPh_3_)_4_ (46 mg, 0.04 mmol), THF(30 mL) and 2M K_2_CO_3_ (1 mL) were added in a 100 mL flask under N_2_ atmosphere. The reaction solution was refluxed for 12 h. The resulting mixture was extracted with DCM and washed with distilled water. The organic layer was dried with anhydrous NaSO_4_. The solvent was removed under vacuum, and the crude compound was purified by column chromatography with PE:DCM = 1:1 as the eluent to give compound **3c** (yield: 34%, 121 mg). ^1^H NMR (CDCl_3_, 500 MHz, ppm): 10.02 (s, 1H), 8.82 (d, *J* = 1.65 Hz, 1H), 8.26 (d, *J* = 4.35 Hz, 1H), 8.21 (d, *J* = 8.01 Hz, 1H), 8.16–8.11 (m, 2H), 8.07 (s, 1H), 8.04 (s, 1H), 7.92 (d, *J* = 7.48 Hz, 1H), 7.87 (d, *J* = 4.04 Hz, 1H), 7.55 (d, *J* = 8.70 Hz, 1H), 7.51–7.47 (m, 1H), 7.41 (d, *J* = 8.21 Hz, 1H), 7.23 (d, *J* = 7.21 Hz, 1H), 4.38–4.25 (m, 4H), 2.27–2.11 (m, 2H), 1.41–1.16 (m, 16H), 0.90–0.81 (m, 6H), 0.81–0.76 (m, 6H). ^13^C NMR (CDCl_3_, 125 MHz, ppm): 183.11, 154.34, 152.83, 149.27, 143.10, 142.45, 142.19, 136.93, 136.84, 136.38, 127.82, 127.71, 127.12, 126.73, 125.82, 123.70, 123.37, 123.70, 122.72, 121.14, 120.13, 118.02, 108.84, 99.27, 47.87, 39.33, 31.08, 28.86, 24.56, 23.11, 14.12, 11.03. HRMS (ESI, m/z): [M]^+^ Calcd. for (C_45_H_48_N_4_OS_2_), 724.3270, found: 724.3280.


**(E)-3-(5-(5,11-bis(2-ethylhexyl)-5,11-dihydroindolo[3,2-b]carbazol-2-yl)thiophen-2-yl)-2-cyanoacrylic acid (D6)**


Compound **3a** (100 mg, 0.17 mmol), cyanoacetic acid (43 mg, 0.75 mmol), piperidine (101 g, 1.18 mmol) and chloroform (30 mL) were added in a 100 mL flask under N_2_ atmosphere. The reaction solution was refluxed for 12 h. After that, the resulting mixture was acidified with 2 M HCl aqueous solution (40 mL) for 0.5 h. The resulting mixture was extracted with DCM and washed with distilled water. The organic layer was dried with anhydrous NaSO_4_. The solvent was removed under vacuum, and the crude compound was purified by column chromatography with MeOH:DCM = 1:10 as the eluent to give dye **D6** (yield: 80%, 89 mg). ^1^H NMR (CDCl_3_, 500 MHz, ppm): 8.72 (s, 1H), 8.46 (s, 1H), 8.42 (d, *J* = 5.2 Hz, 1H), 8.29 (d, *J* = 7.5 Hz, 2H), 8.03 (d, *J* = 3.8 Hz, 1H), 7.88 (d, *J* = 8.0 Hz, 1H), 7.79–7.78 (m, 1H), 7.63–7.60 (m, 1H), 7.53 (d, *J* = 8.0 Hz, 1H), 7.48–7.46 (m, 1H), 7.22–7.18 (m, 1H), 4.40–4.32 (m, 4H), 2.18–2.08 (m, 2H), 1.41–1.26 (m, 16H), 0.91–0.86 (m, 6H), 0.80–0.75 (m, 6H). ^13^C NMR (CDCl_3_, 125 MHz, ppm): 164.41, 155.85, 142.66, 142.15, 136.82, 133.73, 126.41, 124.09, 123.45, 123.06, 122.49, 122.23, 120.94, 118.82, 110.34, 109.5, 100.60, 100.40, 47.31, 38.91, 30.51, 28.56, 24.32, 23.31, 14.20, 11.00. HRMS (ESI, m/z): [M − H]^−^ Calcd. for (C_42_H_47_N_3_O2S), 656.3312, found: 657.3303.


**methyl 4-(7-(5,11-bis(2-ethylhexyl)-5,11-dihydroindolo[3,2-b]carbazol-2-yl)benzo[c][1,2,5]thiadiazol-4-yl)benzoate (D7)**


Compound **3b** (100 mg, 0.13 mmol), NaOH aqueous solution (9.5 mL, 20%), MeOH (15 mL) and THF (30 mL) were added in a 100 mL flask. After that, the resulting mixture was acidified with 2 M HCl aqueous solution (40 mL) for 0.5 h. The resulting mixture was extracted with DCM and washed with distilled water. The organic layer was dried with anhydrous NaSO_4_. The solvent was removed under vacuum, and the crude compound was purified by column chromatography with MeOH:DCM = 1:10 as the eluent to give dye **D7** (yield: 75%, 74 mg). ^1^H NMR (CDCl_3_, 500 MHz, ppm): 8.68 (s, 1H), 8.25–8.21 (m, 3H), 8.06 (d, *J* = 8.20 Hz, 3H), 8.01 (s, 1H), 7.95 (s, 1H), 7.78–7.70 (m, 2H), 7.53–7.49 (m, 1H), 7.43–7.36 (m, 2H), 7.29–7.26 (m, 1H), 4.31–4.01 (m, 4H), 2.28–2.09 (m, 2H), 1.49–1.26 (m, 16H), 1.01–0.96 (m, 6H), 0.92–0.87 (m, 6H). ^13^C NMR (CDCl_3_, 125 MHz, ppm): 171.92, 154.41, 153.83, 142.76, 142.18, 136.83, 136.66, 135.48, 130.41, 130.30, 129.13, 128.87, 128.51, 127.14, 126.91, 125.74, 123.16, 122.91, 122.80, 121.00, 120.11, 117.98, 108.67, 99.16, 47.69, 39.32, 31.08, 28.87, 24.50, 23.09, 14.14, 11.03. HRMS (ESI, m/z): [M − H]^−^ Calcd. for (C_47_H_50_N_4_O_2_S), 733.3582, found: 733.3575.


**(E)-3-(5-(7-(5,11-bis(2-ethylhexyl)-5,11-dihydroindolo[3,2-b]carbazol-2-yl)benzo[c][1,2,5]thiadiazol-4-yl)thiophen-2-yl)-2-cyanoacrylic acid (D8)**


The synthetic method of the dye **D8** was similar with that of the compound **D6**, and the residue was isolated by silica gel column chromatography with MeOH:DCM = 1:8 as the eluent to give orange solid dye D8 (yield: 78%, 85 mg). ^1^H NMR (DMSO, 500 MHz, ppm): 8.88 (s, 1H), 8.33–8.27 (m, 3H), 8.25–8.22 (m, 2H), 8.18 (s, 1H), 8.12 (d, *J* = 8.52 Hz, 1H), 7.99 (d, *J* = 7.52 Hz, 1H), 7.83 (d, *J* = 3.76 Hz, 1H), 7.57–7.53 (m, 2H), 7.49–7.46 (m, 1H), 7.22–7.19 (m, 1H), 4.36–4.29 (m, 4H), 2.18–2.11 (m, 2H), 1.41–1.16 (m, 16H), 0.90–0.81 (m, 3H), 0.81–0.76 (m, 9H). ^13^C NMR (DMSO, 125 MHz, ppm): 164.31, 154.34, 152.56, 150.61, 149.11, 143.45, 142.03, 138.59, 136.74, 136.62, 131.89, 131.16, 130.11, 128.01, 127.41, 126.81, 126.28, 125.71, 123.71, 122.98, 122.65, 118.31, 117.18, 115.94, 109.41, 100.15, 47.27, 38.99, 30.51, 28.40, 24.32, 22.98, 14.35, 11.24. HRMS (ESI, m/z): [M]^+^ Calcd. for (C_48_H_49_N_5_O_2_S_2_), 791.3322, found: 791.3340.

## 3. Results and Discussion

### 3.1. Photophysical Properties

The UV–Visible absorption spectra of dyes **D6**, **D7** and **D8** in DCM solution with concentration of 1 × 10^−5^ M are presented in Figure 2, and the corresponding data are listed in Table 1. The shorter wavelengths located at 300–370 nm are ascribed to the localized π–π* transitions of the conjugated backbone, while the band of 400–600 nm is attributed to the intramolecular charge transfer (ICT) transition from donor to acceptor [30,31]. The maximum absorption wavelength (*λ_max_*) for dyes **D6**, **D7** and **D8** are 500 nm, 463 nm and 529 nm, respectively. Dyes **D6**–**D8** show molar extinction coefficient(ε) with values of around 4.59 × 10^4^ M^−1^cm^−1^, 1.58 × 10^4^ M^−1^cm^−1^ and 1.73 × 10^4^ M^−1^cm^−1^, respectively. The remarkably high ε of dye **D6** is due to the introduction of a thiophene group, which expand the conjugate plane. Therefore, the light absorption properties of dye **D6** have been improved [32,33]. In addition, a larger ε allows a thinner TiO_2_ film, which shortens the electrolyte diffusion distance in the film and, thus, reduces the charge recombination during transport [24]. Furthermore, dye **D8** shows the broadest absorption band of the dyes, and the *λ_max_* red-shifted about 87 nm and 21 nm, in comparison with dyes **D6** and **D7**, respectively, due to its stronger ICT transition.

The normalized absorption spectra of dyes **D6**–**D8** adsorbed on 12 μm TiO_2_ films are shown in Figure 2b. After being anchored onto the TiO_2_ films, the absorption spectra (300–750 nm) of the three dyes were wider, compared with the spectra in solution (300–700 nm), which is beneficial to light-harvesting and J_sc_ enhancement. Compared with the *λ_max_* in solution, the *λ_max_* of dyes **D6**–**D8** on TiO_2_ film are blue-shifted by about 49, 1 and 8 nm, respectively. The blue-shift of absorption on TiO_2_ film is mainly attributed to the deprotonation of carboxylic acid and H-type aggregation [34].

### 3.2. Electrochemical Properties

Electrochemical properties of dyes **D6**–**D8** were studied by cyclic voltammetry (CV) curves in dichloromethane solution, with TBAPF_6_ as the electrolyte. The CV curves are shown in Figure 3a, and the corresponding data are displayed in Table 1. The three dyes show quasar-reversible oxidation waves, which indicate that the electrochemical properties of these dyes are stable. The oxidative potential (E_ox_) versus normal hydrogen electrode (vs. NHE) of dyes **D6**–**D8** are 0.87, 1.01 and 0.81 V, respectively, which are higher than the I^−^/I_3_^−^ potential (+0.4 V), indicating that the oxidized dyes can be regenerated effectively [11,35,36,37]. The energy gaps (E_0-0_) of dyes **D6**–**D8**, calculated according to UV–Vis absorption spectra, were 2.17, 2.36 and 1.95 eV, respectively. Therefore, the reductive potentials of dyes **D6**–**D8**, calculated according to the equation of E_red_[V] = E_ox_ − E_0-0_, were −1.30, −1.35 and −1.08 V (vs. NHE), respectively, which are much higher than the conduction-band edge of TiO_2_ (−0.5 V). This indicates that the driving force for the electron injection from dyes to the TiO_2_ film is sufficient. Figure 3b intuitively illustrates the energy-level distribution of three dyes and the flow direction of electrons in the device.

### 3.3. Theoretical Calculations

The structures of dyes **D6**–**D8** have been further analyzed by using (B3LYP/6-31G (d, p) level). The electron distributions in the highest occupied molecular orbital (HOMOs) and lowest unoccupied molecular orbital (LUMOs) of dyes **D6**–**D8** are shown in Table 2. The HOMOs of dyes **D6**–**D8** reside mostly on the indolo[3,2-b] carbazole, and the LUMOs reside over the thiophene, benzothiadiazole and cyanoacrylic acid acceptor. We can observe the significant overlapping between the HOMOs (indolo[3,2-b] carbazole) and LUMOs (thiophene and benzothiadiazole), which facilitates charge transfer transition from the donor to the acceptor. It is worth noting that the electron density of the excited state of dye **D7** is not completely localized at the anchoring units. This unoptimal electronic matching result may influence the electron injection from dye **D7** to the TiO_2_ and, thus, cause the unexpectedly lower J_sc_ recorded. The optimized ground-state geometries (Table 3) indicate that the indolo[3,2-b] carbazole unit presents a rigid planar structure. The dihedral angles (between indolo[3,2-b] carbazole to acceptor) of dyes **D6**–**D8** are 21.8°, 36.0° and 32.7°, respectively. The dihedral angle between thiophene and phenyl (21.8° for dye **D6** and 0.7° for dye **D8**) is smaller than that between two phenyls (36° for dye **D7** and 32.7° for dye **D8**). This indicates that thiophene group can improve planarity and conjugation, which is good for dye ICT transition [38,39,40].

### 3.4. Photovoltaic Properties

The photovoltaic properties of dyes **D6**–**D8** and **N719** are performed under standard conditions (AM 1.5, 100 mW cm^−2^). The photocurrent–voltage (J–V) curves are shown in Figure 4a, and the corresponding data of short circuit photocurrent density (J_sc_), open-circuit voltage (V_oc_), fill factor (FF) and power conversion efficiency (PCE) are shown in Table 4. In the UV–Visible absorption spectra, dye **D8** shows a broader absorption band, which may lead to a higher J_sc_. However, the J_sc_ of the DSSCs based on dyes **D6**–**D8** are 12.55, 10.76 and 10.17 mA cm^−2^, respectively. It is clear that the dye **D6**-based cell shows higher J_sc_ than that of dyes **D7** and **D8**, which is due to its higher molar extinction coefficients. Furthermore, the dye loading amount of dyes **D6**–**D8** are 4.16 × 10^−7^ mol cm^−2^, 2.81 × 10^−7^ mol cm^−2^ and 2.51 × 10^−7^ mol cm^−2^, respectively, which is consistent with the J_sc_ of the three dyes. The V_oc_ of the DSSC based on dyes **D6**–**D8** is 745, 744 and 668 mV, respectively. Compared with the DSSCs of dyes **D7** and **D8**, dye **D6**-based DSSC shows the highest PCE of 5.41%, due to its high J_sc_. Finally, these three dyes exhibit good power conversion efficiency, which indicates that indolo[3,2-b]carbazole group might be a good donor.

The incident photon-to-current conversion efficiency (IPCE) spectra for the cells is shown in Figure 4b. As can be seen, all the three dyes exhibit broad response which indicates that dyes **D6**–**D8** can convert the visible light to photocurrent efficiently. The maximum spectra response band of dyes **D6**–**D8** are at 710, 660 and 770 nm, respectively. The absorption range of dye **D8**-based DSSC is obviously broader than that of dyes **D6** and **D7**, due to the incorporated benzothiadiazole-thiophene unit decreasing the E_0-0_. However, dye **D6** shows strong response in the range from 350 to 550 nm, with a highest IPCE value of 72.55% at 480 nm. Dyes **D7** and **D8** display the maximum IPCE values of 68.4% at 500 nm and 47.74% at 525 nm, respectively. Therefore, the best IPCE performance is observed on the dye **D6**, among the three dyes. It mainly originates from its much higher molar extinction coefficient and more dye loading amount. These results are also in good accordance with the J_sc_ value obtained in above J–V measurements.

The chenodeoxycholic acid (CDCA) co-sensitization was tested by blending dyes **D6**, **D7** and **D8** with different ratios of CDCA in the THF dye bath, respectively. The corresponding curve and data are shown in Figure 5 and Table 4, respectively. Generally, the PCE of most dyes can be improved by the addition of CDCA, because it can restrain the dye aggregation. However, for aggregation-free organic dyes, the addition of CDCA would reduce adsorption of the dye molecules on the TiO_2_ film, which would reduce the short-circuit current density and, hence, the power conversion efficiency [41]. The photovoltaic properties of the DSSCs based on dyes **D6**–**D8** with no additives and with CDCA as co-sensitizer were investigated. As can be seen in Figure 5a,b, the decrease of J_sc_ is found in dyes **D6**- and **D7**-based DSSC after the co-sensitization. When the ratio of CDCA increases, both the J_sc_ and PCE of these two dyes decrease. It can be explained that indolo[3,2-b]carbazole and its long alkyl chain can restrain dye aggregation efficiently. Meanwhile, adopting CDCA as the co-adsorbent is unfavorable for these two dyes adsorptions on TiO_2_ films. However, Figure 5c shows that J_sc_ increases by adding 10 times of CDCA. This indicates that **D8** presents aggregation and improved by the addition of CDCA.

### 3.5. Electrochemical Impedance Spectra (EIS) Analysis

In order to study the charge recombination and charge transfer process in the DSSCs, the electrochemical impedance spectroscopy (EIS) analysis is performed in the dark. The measurement is under a bias of −0.7 V, and the frequency range is 0.1 Hz–10 kHz. The corresponding Nyquist plots are shown in Figure 6a. The first semicircle at higher frequencies corresponds to the electron transport at the Pt/electrolyte interface, and the last one at lower frequencies corresponds to the charge recombination at the TiO_2_/dyes/electrolyte interface. The larger value of charge transfer resistance (R_ct_) reflects the lower charge recombination, smaller dark current and larger open circuit voltage [42,43]. The values of R_ct_ can be deduced by fitting curves using ZView complex nonlinear least-square regression software (AMETEK, Leicester, UK) The order of R_ct_ values is **D8** (63.7 Ω) < **D7** (113.0 Ω) < **D6** (155.5 Ω), which is consistent with the order of V_oc_ values of **D6** (668 mV) < **D7** (744 mV) < **D8** (745 mV). The electron lifetime (τ) of the dyes **D6**–**D8** can be calculated from the Bode phase plots (Figure 6b) to support the trends of V_oc_. The electron lifetime (τ) is calculated from τ = 1/(2πf) [44,45], where f is the peak frequency in lower frequency. The order of τ is **D6** (25.8 ms) > **D7** (25.5 ms) > **D8** (8.3 ms), which is in line with their V_oc_.

### 3.6. Stability Study

The study of stability of dyes is very important for the DSSCs’ applications. The devices, for long term stability, were encapsulated and stored in dark conditions. The evolution of efficiency was measured by testing the PCE values of three DSSCs, based on dyes **D6**–**D8**, once a day. As we can see in Figure 7, nearly 90% of initial efficiency (*η*_0_) for all three DSSCs remained over 480 h, which suggests that our dyes are stable and, thus, suggesting application potential.

## 4. Conclusions

Three novel metal-free organic dyes (**D6**, **D7** and **D8**) with different acceptors were synthesized and applied in DSSCs. Dye **D6**, containing thiophene cyanoacetic acid acceptor, exhibits much higher ε and dye loading amount than the other two dyes, which indicates that dye **D6** possesses better light-harvesting capability. By the adoption of alkylated indolo[3,2-b]carbazole, dye aggregation was efficiently suppressed. Hence, the power conversion efficiencies of the dyes without co-sensitization with CDCA are better, as compared with the co-sensitized devices. Because of the introduction of benzothiadiazole thiophene group, dye **D8** shows the broadest light-response range (300–770 nm) in IPCE curve, due to the decrease of the E_0-0_. However, its lower ε and dye loading amount leads to poor photovoltaic performance. The DSSC-based dyes **D6**–**D8** show over 90% of initial efficiency during 480 h. Finally, dye **D6** exhibits the best performance among these dyes, with the highest power conversion efficiency of 5.41%, mainly due to its highest J_sc_ (12.55 mA cm^−2^).

## Data Availability

Data sharing is not applicable to this article.

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
