# Peer review of "Novel Red Light-Absorbing Organic Dyes Based on Indolo[3,2-b]carbazole as the Donor Applied in Co-Sensitizer-Free Dye-Sensitized Solar Cells"

_materials, 2021, doi:10.3390/ma14071716_

Round 1

Reviewer 1 Report

The paper Xiao et al. deals with the synthesis of novel carbazole-based dyes to be implemented as sensitizer in Dye-Sensitized Solar Cells. The results, even quite far from literature records, is worth of publication. Yet, before this steps some issues should be addressed (listed below). Indeed, the reported results are not sufficiently discussed and their justification is weak in some points. 

1) Expecially in the introduction, references are not completely up to date. Some recent reviews/papers dealing with the carbazole-based dyes and, more generally DSSCs, should be insert. (Some suggestion are: 10.1016/j.molliq.2020.113189; 10.1021/acsami.0c14952.; 10.1039/D0GC01148G; 10.1063/5.0017041)

2) The ratio behind the design of the sensitizers should be better explained in the manuscript

3) NMR and Mass Spectra should be presented as supporting information to both readers and reviewers.

4) Line 282. Why the insertion of the thiophene is beneficial for a higher molar extintion coefficient? Is it due to a better conjugation due to increased planarity?

5) Line 291. The sentence "It is noteworthy that the absorption peaks of D6-D8 are blue-shift by about 49, 1 and 8nm, respectively" is misleading being the shift of 49 nm referred to a different peak. Please clarify it in the text.

6) CV analyses should be revised (at least for compound D7). Indeed, usually the oxidation potential is calcualted as the semisum of the Eox,1 and Ered,1 (oxidazion and reduction related to the same peak). In the present case, the election peak should be the one located at lower potential (see figure 3a). Yet, from the figure is clear that the Eox of D7 is shifted toward higher potential (I would guess 1.05-1.10 vs NHE). The value given by the authors (i.e. 0.83) is clearly wrong and it also reflects on the LUMO value. Please amended this section.

7) Line 312. The authors stated "Compared to D6 and D7, D8 has weaker driving force for the electron injection from dyes to TiO2 film and it may be unfavorable for obtaining high short circuit current densities." This is not completely true; indeed, a driving force of 0.2 eV is enough to assure a quantitative injection to the CB of TiO2. Please correct. 

8) With respect to theoretical calculations, the electron density of the excited state of D7 is not completely localized at the anchoring units (i.e. close to the TiO2 surface). The unoptimal electronic matching resulting from this could be the reason behind the unexpectedly lower Jsc recoreded. Could the authors comment on that?

9) With respect to PV characterization is not clear if the higher Jsc powered by D6 is due to higher ε (chemical reason) or better dye-laoding (physical reason). This review would guess the second one being the ε of all dyes sufficiently high to assure a quantitative light harvesting (APCE measurements coulf help in clarify this point). Additionally, it is not clear the reason behind the lower Voc recorded for D8. 

10) IPCE data should also present the integrated current value

11) With respect to the CDCA employment, this review would suggest additional experiment. First of all, CDCA/dye ratio 3:1 is remarkably low (see 10.1016/j.orgel.2013.06.026) and it would results in unsufficient CDCA loading. Secondly, in order to prove the unnecessity of CDCA, different CDCA/dye ratio should be invesitgated (see as example 10.1149/2.0971714jes). 

12) EIS analyses is quite unclear to this reviewer. Why the auhtors biased the device at -0.7 V if the perform measurements under dark? Usually, bias is required to counterbalanced the potential difference generated by light irradiation and to have the lowest charge possible flowing in the cell during the measurements. In dark, the Voc of the device is nearly zero and the bias would induce a charge flow that seriously undermine the validity of the obtained results. Please check this (this reviewer suggest to afford new EIS analyses).

12) Long term stability of device is really important, yet the analyses made by the authors lacks for details (storage conditions, electrolyte refilling...). Moreover, the shape of the ageing curve have a (particular) sinousoidal shape.

Reviewer 2 Report

The manuscript "Novel red light-absorbing organic dyes based on indolo[3,2-b]carbazole as the donor applied in co-sensitizer free dye-sensitized solar cells" by Zhanhai Xiao, Bing Chen, and Xudong Cheng  describes the synthesis of three new compounds that were characterized electrochemically and spectroscopically, as well as tested as active components in dye-sensitized solar cells. The topic should be quite interesting to the readership of the journal Materials, however there are still some open questions and concerns that need to be addressed, before publication may happen:

1. In line 59, the authors claim that narrow band gaps are detrimental for DSSC applications, however in lines 64-66 they describe how the LUMO can be lowered, yielding a reduced band gaps, which in this context seems to be something positive (specifically if the following sentence in lines 66-68 is considered). Therefore, the authors should rework this section to ensure that there are seemingly no contradictions like these.

2. The bottom part of the reaction in Scheme 1 is not very well explained and it is not clear how the different compounds are synthesized from this image. Therefore, a re-design might be necessary for this part of Scheme 1.

3. In Table 1, footnote [b], the authors describe how E_0-0 (i.e. the optical band gap) was determined. Have the authors used a Tauc plot, as should be done (DOI: 10.1016/0025-5408(68)90023-8)? The authors should then describe the determination of the band gap in more detail.

4. In lines 332-333, the authors mention the role of the dihedral angle and the subsequent changes in the energetic landscape. This should be expanded a bit more and also some references should be added that discuss this topic in other organic compounds used for photophysics (DOI: 10.1039/D0TC02136A; 10.1016/j.dyepig.2020.108540).

5. In Fig. 5, the IPCEs are shown. Could an integration of the IPCE yield a theoretical maximum J_SC, where no transport losses are included? Something to this end was mentioned in the text, but a calculation and listing of these idealized J_SCs might be warranted.

6. In line 384, the authors mentioned a forward bias of -0.7V. How can a forward bias be negative?

7. In Fig. 6, the authors show the results from their impedance analysis. Could the authors also add the equivalent circuit models they use (as an inset in Fig. 6)?

8. Some of the Figures and Schemes, specifically the ones containing chemical structures would benefit from higher resolution images.

9. Spelling errors, typos, and other minor comments are shown in the attached PDF.

Round 2

Reviewer 1 Report

Authors kindly replied to all referees' comments. 

The manuscript could now be published in Materials